# RC2020 Report: Learning De-biased Representations with Biased Representations

Anonymous

## Reproducibility Summary

**Scope of Reproducibility**

The authors formalize and attempt to tackle the so called "cross bias generalization" problem with a new approach they introduce called ReBias[1]. This report contains results of our attempts at reproducing the work in the application area of Image Recognition, specifically on the datasets biased MNIST and ImageNet. We compare ReBias with other methods - Vanilla, Biased, RUBi[2] (as implemented by the authors), and conclude with a discussion the validity of the claims made by the paper. Our reproducibility source code is available in the supplementary material we've uploaded with the revision.

**Methodology**

We used the authors' code available (*source*) and did not modify any part of it to make sure the initial conditions of the experiments were identical. We did not re-implement any part of the pipeline except for ImageNet, implementation issues of which we discuss later on. The total training time, with final results averaged over 3 runs, amounted to 128 hours on on a single NVIDIA GeForce GTX 1080 8GB GDDR5X with 2560 CUDA cores.

**Results**

We were able to reproduce results reported for the biased MNIST dataset to within 1% of the original values reported in the paper. Like the authors, we report results averaged over 3 runs. However, in a later section, we provide some additional results that appear to weaken the central claim of the paper. We were not able to reproduce results for ImageNet as in the original paper, but present our results and a further discussion.

**What was easy**

We found the reproducibility for the *biased MNIST* dataset to be easy. The code provided was clear to understand and ready to run as a simple terminal command.

**What was difficult**

Although the majority code used by the authors for the training pipleline was made publicly available, there were a few things we had to be careful about, including using the right version of the library *torchvision* (use latest version, version used by the authors is different), using the right WNIDs for scraping the *9-class ImageNet* dataset (actual dataset used in training is not provided). We found it difficult to reproduce two of the methods ReBias is compared against (*HEX*[3] and *Learned-Mixin*[4]), as they were the authors' implementation of the methods (authors' implementation of *HEX* is not available). The most significant difficulty faced was attempting to avoid exploding gradients on one of the experimental settings, even though we used the exact same training pipeline.

**Communication with original authors**

We had some preliminary contact with the authors regarding the implementation of the methods discussed above. We were informed that the HEX implementation was exactly as in the original work, and that the implementations for RUBi and Learned-Mixin' had been adapted from the NLP domain into the vision domain. Further, the authors confirmed some details about the dataset and training which were previously not clear.

Preprint. Under review.

# 1 Introduction

The paper investigates the problem of "cross bias generalization", which is to reduce the impact of bias cues CNN models use to predict outcomes of image inputs. These bias shortcuts ultimately hamper the model's ability to generalize well to unseen inputs, as, for example, images of a frog in a swamp may be quite frequent in a training dataset, but an image of a frog in a living room may be rare. Testing on this example, however, a model strongly correlated with the bias of an object (in this case, a swamp) will perform poorly on this test image, where the bias has shifted. A different background (bias) does not change the nature of the object in question. Given a signal cue $S$ for an image input $X$, the true output $Y$ [$Y^{tr}$ and $Y^{te}$ being the training and test distributions], and the associated bias of $Y$ as $B$ [$B^{tr}$ and $B^{te}$ being the biases in the training and test distributions], the cross bias problem can be summarized like so:

- $p(B^{tr}) \not\perp\!\!\!\perp p(Y^{tr})$
- $p(B^{tr}, Y^{tr}) \neq p(B^{te}, Y^{te})$
- $p(B^{tr}) = p(B^{te})$

As an example, the training data may contain two types of images, ($Y^{tr} = frog, B^{tr} = swamp$), and ($Y^{tr} = bird, B^{tr} = sky$), but the test set contains unusual combinations of the classes and biases: ($Y^{tr} = frog, B^{tr} = sky$), ($Y^{tr} = bird, B^{tr} = swamp$). **In essence, we have a crossover of the biases**. A biased model will have trouble discriminating the classes, as we will see in the results later.

There are primarily three ways one can improve cross bias generalization. Firstly, if one could disentangle bias $B$ from signal $S$ completely, then one could eliminate the bias dimension in their dataset and train bias free on this normalised data. However, as [1] show, for texture biases in CNNs, this is an unrealistic task. Secondly, if one could have a data generation procedure $p(X|B)$, one could de-bias a training dataset. But this means, in our frog example, we may need to collect images of frogs with varied backgrounds [2]. This, again, is unrealistic, not to mention expensive. The third approach is the reverse of the second. If one could design a predictive algorithm for $p(B|X)$, which means one can describe the bias for every possible input $X$, it is possible to de-bias the training data [3, 4]. This is also an issue as for biases like textures, it is impossible to enumerate all possible textures and associate each with a label.

**However, the third approach can be useful with certain assumptions. These assumptions are what lead to ReBias.** In section 2 we briefly summarise the intuition behind ReBias. In section 3 we point out the claims made in the the paper that we wish to verify. In section 4 we describe our replication methodology and in section 5 we present and discuss our results. Subsequently in section 6 we analyze the claims made by the authors and finally, section 7 summarizes this report.

# 2 ReBias

Although it is computationally intractable to solve for $p(B|X)$ deterministically, one could approximate it using different bias models $G$. For image recognition, and in the context of CNNs, [5] shows that receptive fields are important factors in capturing texture biases. Their empirical evidence suggests that **smaller filter sizes lead to increased texture bias in CNN architectures**. In this case, $G$ would simply be a set of models with smaller receptive fields. And learned model $g \in G$ would then make predictions on images based on texture biased cues. They are more likely to overfit to texture biases.

For a given bias-signal pair $(B, S)$, G is called a **bias characterizing model class**, if for every possible joint distribution $p(B, X)$ (where $X$ is the input signal), the following two conditions satisfy:

- $\exists g \in G, p(B|X) \approx g(X)$
- $\forall g \in G, g(X) \perp\!\!\!\perp S|B$

Depending on the application task, the set $G$ would vary accordingly. For image recognition, as discussed, $G$ refers to models with smaller filter sizes (leading to increased texture bias). For action recognition tasks, $G$ can be chosen to be a set of 2D CNN models, etc.

This finally brings us to the proposed method: ReBias.

Here the idea is to design a model $p(Y|X)$ that is encouraged to be maximally independent of $g(X) : g \in G$, the set of all possible biased models with smaller receptive fields. In short, one intentionally overfits a collection of biased models on the data, and then encourages $f(X)$ to be as independent as possible from $G$. To quantify this metric, the authors use the **Hilbert-Schmidt Independence Criterion (HSIC)** [6]. The intuition for using this metric is that for RBF kernels $k, l$, and for two random variables $U, V$, $HSIC^{k,l}(U, V) = 0$ if and only if $U \perp\!\!\!\perp V$. This is exactly what is needed with respect to the random variables in the authors' case: $f(X)$ and $G$. Thus:

$$HSIC(f, G) := \max_{g \in G} HSIC(f, g)$$

This allows the authors to formulate the ReBias loss function:

$$\min_f [L(f, X, y) + \lambda \cdot \max_{g \in G} HSIC(f, g)]$$

It can be observed that this is a min-max optimisation procedure, with the HSIC criterion added on top of a standard loss function commonly used in deep learning (categorical crossentropy, in this case). This is achieved by two successive update steps, first the *max* operation, and then the overall standard minimisation of the loss over $f$.

This sums up ReBias. Create one standard convolutional model $f$, and a set of bias models $G$ with smaller kernel (filter) sizes than $f$. Optimise following the procedure above. The intuition is that, as one set of models continuously overfit to the data, the main model is encouraged to stay independent on this set, and thus independent from texture biases learnt by $G$ during training.

# 3   Scope of reproducibility

From the previous sections, we thus proceed to verify the following claims in the paper:

- Claim 1: The min-max procedure for ReBias converges.

- Claim 2: The deliberately texture biased model g helps $f$ reduce cross bias.

- Claim 3: The Hilbert-Schmidt Independence Criterion (HSIC) utilized in the min-max objective function of ReBias is useful in reducing cross bias.

- Claim 4: ReBias solves the cross-bias generalization problem, by robustly improving a biased model's cross-bias test accuracy.

# 4   Methodology

The authors of ReBias [1] provide a code (*source*) for their implementation containing the entire training and validation pipeline, for which they provide a fairly detailed documentation. We did not make major modifications to it as there are quite a few areas where slight tweaks may lead to incorrect reproduction. However there were certain parts of the pipeline that we needed to implement on our own, and modify some specific parts of the code because of certain issues we faced when using the code provided by the authors. We elaborate this further in section 5. The GPUs we used for our runs were 1xNVIDIA GeForce GTX 1080 8GB GDDR5X with 2560 CUDA cores. We provide the shells scripts used and the modified bits of code as part of the supplementary material.

## 4.1   Experimental setup and descriptions

Algorithms 1,2,3 and 4 shown below, demonstrate the training logic (as deduced from the code provided by the authors) for the ReBias, Vanilla, Biased and RUBi models.

---
**Algorithm 1** ReBias

**repeat**
    **for each** mini-batch samples $x, y$ **do**
        update $f$ by solving $\underset{f \in F}{\mathrm{argmin}}\ \mathcal{L}(f(x), y) + \lambda HSIC(f(x), g(x))$
        update $g$ by solving $\underset{g \in G}{\mathrm{argmin}}\ \mathcal{L}(g(x), y) - \lambda_g HSIC(f(x), g(x))$
    **end for**
**until** converge

---

---
**Algorithm 2** Vanilla

**repeat**
    **for each** mini-batch samples $x, y$ **do**
        update $f$ by solving $\underset{f \in F}{\mathrm{argmin}}\ \mathcal{L}(f(x), y)$
    **end for**
**until** converge

---

---
**Algorithm 3** Biased

**repeat**
    **for each** mini-batch samples $x, y$ **do**
        update $g$ by solving $\underset{g \in G}{\mathrm{argmin}}\ \mathcal{L}(g(x), y)$
    **end for**
**until** converge

---

---
**Algorithm 4** RUBi

**repeat**
    **for each** mini-batch samples $x, y$ **do**
        update $f$ by solving $\underset{f \in F}{\mathrm{argmin}}\ \mathcal{L}(f(x), y) + \lambda RUBi(f(x), g, y)$
        update $g$ by solving $\underset{g \in G}{\mathrm{argmin}}\ \mathcal{L}(g(x), y)$
    **end for**
**until** converge

Note: $f$ and $g$ are trained simultaneously but separately. The function $RUBi$, implemented by the authors of ReBias uses features from the biased network ($g$), the prediction of the target network $f(x)$ and the ground truth labels ($y$).

---

Here, $f$ is the original model which needs to be de-biased, $g$ is the texture biased model with low receptive field, $\mathcal{L}(f(x), y)$ is the loss function (like cross-entropy loss), $HSIC(f(x), g(x))$ is the Hilbert-Schmidt-Independence-Criterion, $\lambda$ and $\lambda_g$ are hyper-parameters.

In our experiments we repeated the experiments three times, reporting the average of 3 runs (like the authors) in the converging cases.

### 4.2 Datasets

For the Image Recognition task, two experiments were conducted: One on **Biased MNIST**, a synthetic dataset created by the authors, and **9-class ImageNet**.

**Biased MNIST**

The authors modify the MNIST [7] dataset by injecting coloured backgrounds to the digits. They select 10 distinct colours for each digit. To create the training set, with probability $\rho$, the pre-selected colour is added to the specific digit, and any other colour with probability $1 - \rho$. Thus, for $\rho = 1$, we have complete bias, and for $\rho = 0$, we have a completely unbiased dataset. The effect of $\rho$ on the distribution of $P(B|S)$ in shown in figure 1.

In this experimental setup, the authors validate the models on a "Biased" validation set (with the same $\rho$ values as the training set) and an "Unbiased" validation set (where $\rho = 0.1$). Note that the "Unbiased" validation accuracy here relates to how robust the model is to cross-biased testing. The idea is that in the Biased dataset, since colours will be strongly correlated to the digits, models suffering from texture bias (the set $G$) will perform strongly (as it picks up the colour cues).

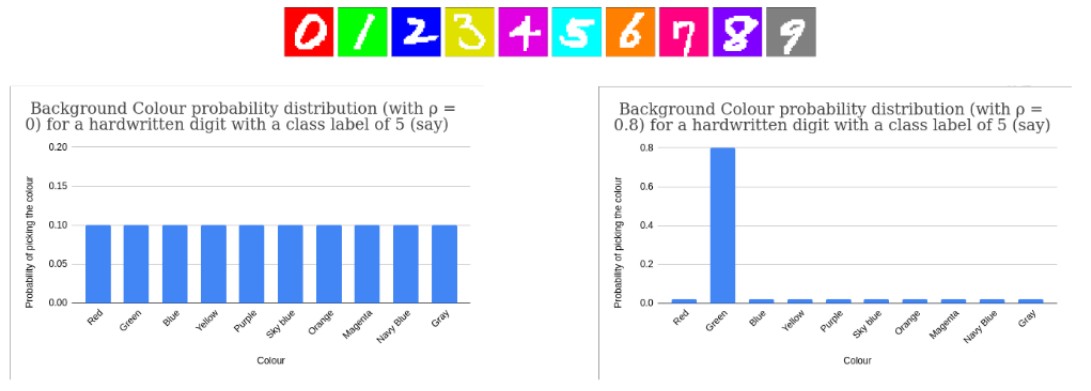

Figure 1: MNIST with colour assignments to digits with a high and low $\rho$

**9-class ImageNet**

The paper uses '9 class ImageNet', a subset of the ImageNet [8] dataset for training, and reports validation accuracies over the '9 class ImageNet' and 'ImageNet-A' [9] and 'ImageNet-C' [10]. ImageNet-A consists of adversarial examples over the ImageNet dataset where top performing ImageNet models fail to correctly classify because they make inferences from some background cues. ImageNet-C consists of manually corrupted images; with 15 corruption types including "noise", "blur", "weather", and "digital" with five degrees of severity. **An improved performance on ImageNet-A and ImageNet-C would indicate that the model learns beyond the bias shortcuts.**

Apart from computing the accuracy over the validation set ('Biased Accuracy'), ImageNet-A and ImageNet-C, they also present an 'unbiased accuracy' - For doing this they propose an unsupervised texture feature clustering technique which inputs each image of the ImageNet validation set into one of 9 cluster-ids (using gram matrices of low-layer feature maps as the texture features of the image, followed by k-means clustering). Then, assuming all images within a cluster have the same texture features (or no variation in texture), they separately compute the proportion of correctly classified images within each of the 9 texture clusters, and then report the average. Note that the paper does not conclusively show that there are no bias variations within the clusters themselves (i.e. why cross bias may still not exist in each of these clusters). Also, the choice of using specifically 9 clusters is not elaborated.

**Availability of the datasets**

The Biased-MNIST dataset was constructed by the authors and is provided in their code base

The ImageNet-A and ImageNet-C datasets are openly available, and the code for computing the 'Unbiased Accuracy' using texture feature clustering is publicly available. However, the exact subset of ImageNet used to build the 9-class ImageNet dataset used by the authors in this paper was not provided. After confirming with the authors we constructed this dataset ourselves, by using the same WNIDs of ImageNet (ILSVRC-2017) sub-classes.

### 4.3 Model architecture and hyperparameters

For the main results shown here we do not change the model architecture or the hyperparameters (except in 9-class ImageNet - explained later)

**Biased MNIST**

The model $f$ is a fully convolutional neural network with 4 layers and kernel sizes of 7 x 7, while the model $g$ has the same architecture except its kernel size is 1 x 1. In both cases, the final convolutional layer is followed by global average pooling, and a linear classifier for predicting the labels (fully connected).

The authors use batch normalization, ReLU as their activation function, 256 as their batch size, a learning rate of 0.001 with a decay factor of 0.1 every 20 epochs, and train for a total of 80 epochs. The authors use ADAM and ADAMP as their optimisers. We present the results with ADAM.

**9-class ImageNet**

In the ImageNet experiment, the model $f$ uses a ResNet18 [11] architecture and the mode $g$ uses a BagNet18 [12] architecture.

The rest of the hyperparameters are the same except for the batch size. The authors use a batch size of 128, however given that we use GPUs much lower memory, we were forced to reduce the batch size. In our experiments we use a batch size of 16. In the next section (5) we show that the ReBias training method results in exploding gradients. While attempting to solve this we made some changes, which we justify and discuss in section 5.2.

# 5 Results

In the following subsections we talk about our results on the Biased MNIST and 9-class ImageNet experimental setup.

## 5.1 Biased MNIST

### 5.1.1 Reproduced Results [Average over 3 runs]

We first present our reproduced results, averaged over 3 runs.

| $\rho$ | Biased Validation Set | | | |
|---|---|---|---|---|
| | Vanilla | Biased | RUBi | ReBias |
| 0.999 | 100 | 100 | 100 | 100 |
| 0.997 | 100 | 100 | 100 | 100 |
| 0.995 | 100 | 100 | 100 | 100 |
| 0.990 | 100 | 100 | 100 | 100 |

Table 1: Validation Accuracy on Biased Data

| $\rho$ | Unbiased Validation Set | | | |
|---|---|---|---|---|
| | Vanilla | Biased | RUBi | ReBias |
| 0.999 | 10.17 | 10 | 10.36 | **21.11** |
| 0.997 | 48.55 | 10 | 35.49 | **61.99** |
| 0.995 | 73.04 | 10 | 65.81 | **75.79** |
| 0.990 | **88.05** | 10 | 84.41 | 87.61 |

Table 2: Validation Accuracy on Unbiased Data

**These results are within 1% of the original results presented in the paper**. As was expected all models perform well on the biased validation sets (because the training data was biased the same way). Things get interesting when we discuss the Unbiased dataset results. A glance at Table 2 told us that the trend of Vanilla vs ReBias accuracies was progressively shifting in favour of the Vanilla model as we slightly decrease $\rho$. This hypothesis was confirmed to be true as we tested for lower values of $\rho$, but still making sure the dataset was sufficiently biased. We test for four additional values of $\rho : 0.98, 0.95, 0.9, 0.85$. These are the additional results we discuss next, not present in the original paper.

### 5.1.2 Additional results not present in the original paper

As is clear from Table 4, Vanilla actually beats ReBias for $\rho = [0.90, .99]$ onwards, while RUBi performs the best for $\rho = 0.85$ and is marginally better than Vanilla. If we look at the plot below, we see a clearer picture that the table represents:

From Figure 2 we can see that the bandwidth of $\rho$ values for which ReBias performs better is demarcated by the red line. Over the rest of the $\rho$ inputs, while the training set is still sufficiently biased, Vanilla performs at least as good or better than ReBias. Intuitively trying to understand this raises two practical questions: (1) Is this experiment's set up too simple for a simple vanilla model to be less prone to cross biases? (2) Or is it that ReBias is effective only when the training set is extremely biased and using a vanilla training technique leads to much lower cross-biased accuracy?

| $\rho$ | Biased Validation Set | | | |
|---|---|---|---|---|
| | Vanilla | Biased | RUBi | ReBias |
| 0.999 | 100 | 100 | 100 | 100 |
| 0.997 | 100 | 100 | 100 | 100 |
| 0.995 | 100 | 100 | 100 | 100 |
| 0.990 | 100 | 100 | 100 | 100 |
| 0.980 | **100** | 99.99 | **100** | 99.99 |
| 0.950 | **99.99** | 99.98 | 99.98 | 99.98 |
| 0.900 | 99.94 | 99.89 | **99.97** | 99.94 |
| 0.850 | **99.94** | 99.8 | 99.89 | 99.93 |

Table 3: Validation Accuracy on Biased Data

| $\rho$ | Unbiased Validation Set | | | |
|---|---|---|---|---|
| | Vanilla | Biased | RUBi | ReBias |
| 0.999 | 10.17 | 10 | 10.36 | **21.11** |
| 0.997 | 48.55 | 10 | 35.49 | **61.99** |
| 0.995 | 73.04 | 10 | 65.81 | **75.79** |
| 0.990 | **88.05** | 10 | 84.41 | 87.61 |
| 0.980 | **93.60** | 10 | 92.66 | 93.48 |
| 0.950 | **96.87** | 12.71 | 96.64 | 96.54 |
| 0.900 | **98.09** | 14.34 | 97.89 | 97.90 |
| 0.850 | 98.41 | 15.51 | **98.46** | 98.24 |

Table 4: Validation Accuracy on Unbiased Data

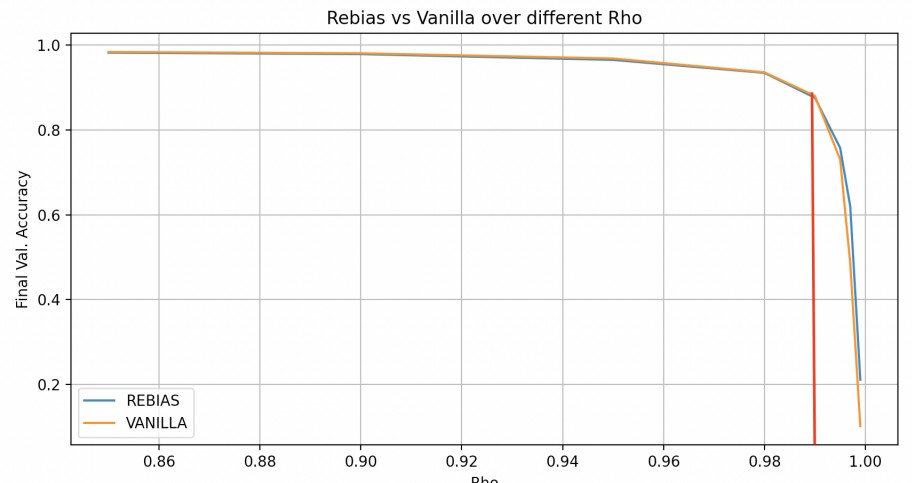

Figure 2: Comparison of Rebias and Vanilla over $\rho \in [0.999, 0.997, 0.995, 0.99, 0.98, 0.95, 0.9, 0.85]$

We answer the first question in the affirmative. The synthetic dataset used in this paper is simplistic as it injects a readily identifiable bias - the color. This allows the definition of the parameter $\rho$, which is accordingly tuned to infer about results. We note here that the conception of $\rho$ would be unclear if one were to use a complicated dataset that is generally texture biased, as opposed to incorporating a single color bias. Examples include Stylized Imagenet, or any other dataset modified by style transfer. However, the authors indeed run their experiments on challenging datasets like Imagenet-A, and as we discuss in a later section, our difficulties (and eventual failure) at reproducing the Imagenet results make it hard to definitively answer the second question we pose. Unfortunately, we can neither prove nor disprove this hypothesis.

## 5.2 ImageNet

We were successfully able to train the Vanilla and the Biased model. However the Rebias model training resulted in exploding gradients, even though we used the same pre-processing and training pipeline. The only hyperparameter that was different from the ones used by the authors was the batch-size (we used a batch size of 16 due to GPU memory constraints). The results we get after using the same hyperparameter setting as the authors (except batch size) is shown in Table 5. As can be seen the results we get after training for 120 epochs are not comparable to the original paper (table 6). What was particularly interesting to us was the fact for the ReBias training type the gradients consistently explode after the very first epoch. Moreover for the biased training type the unbiased validation accuracy reduces to $0\%$ by the end of 120 epochs (worse than a random prediction), which shows that the training procedure for ImageNet completely fails. The fact that this does not happen for our Vanilla training scheme (the only case where there are similar Biased and Unbiased accuracies as the authors, although Imagenet-A accuracies are not similar)on Imagenet, given that we used the same dataset (confirmed by the authors) and pre-processing, provoked us to look into the differences in the training pipeline, i.e. the hyperparameters. These are discussed below.

| Training Type | Validation Types | | |
| --- | --- | --- | --- |
| | Biased | Unbiased | ImageNet-A |
| Vanilla | 88.98 | 88.93 | 5.89 |
| Biased | 79.91 | 0.0 | 7.08 |
| ReBias | 0.0 | 0.0 | 10.39 |

Table 5: Our results on 9-class-ImageNet

| Training Type | Validation Types | | |
| --- | --- | --- | --- |
| | Biased | Unbiased | ImageNet-A |
| Vanilla | 90.8 | 88.8 | 24.9 |
| Biased | 67.7 | 65.9 | 18.8 |
| ReBias | 91.9 | 90.5 | 29.6 |

Table 6: The authors results on 9-class-ImageNet

### 5.2.1 Different hyperparameter settings

1. **Batch Size**: Given that for the Rebias training on ImageNet we got exploding gradients, we tried to artificially increase the batch size from 16 to 128, by accumulating gradients for 8 batches and skipping 8 batches to changing the weights by taking an optimization step - making all the hyperparameters exactly the same as the authors. This however didn't solve the exploding gradients problem or the huge discrepancy in the Unbiased validation accuracy for the Biased training type.

2. **Optimizer and Learning Rate**: Initially, as suggested by the authors, we used the AdamP optimizer with a learining rate of 0.001 and weight decay 0.0001. We later also tried AdamP and Adam optimizer with different learning rates and weight decay. This however did not solve our issue with exploding gradients. However, we note that during these runs the HSIC score fluctuates significantly when tuning the optimizer parameters, leading us to the next set to parameters we tuned in tandem with the ones till now.

3. **HSIC parameters:** The authors implementation of the Hilbert-Schmidt Independence Criterion(HSIC) uses a Radial Basis Function (Rbf) kernel, which has a kernel radius parameter ('sigma') that scales each of the inputs to the HSIC function. The sigmas used can either be fixed or can be changed every epoch. For the Biased MNIST experiment the authors use a fixed sigma, of 1, where as for the ImageNet experiment they used a sigma that changes based on the median of $f(x)$ and $g(x)$, with 25% of the training data chosen randomly. We tried varying these parameters, one of which included the exact same ones used in the Biased-MNIST experiment (since we did not face exploding gradients during this experiment). Unfortunately this also did not fix the issue.

We communicated these issues faced by us in this experiment in detail with the authors, and asked for suggestions or if they had also faced such issues at some point, however we did not receive a response in this short span of time. We did not attempt to use a different network architecture or training/optimization methodology to avoid this exploding gradient problem, since this would lead us to diverge from the reproducibility setting. In this respect, we are forced to finally conclude that our results do not replicate the results presented by the authors - The loss does not converge due to exploding gradients.

## 6 Analysis

From the evidence we get from running our experiments we have a few observations and insights about the claims of the paper.

- *Claim 1: The ReBias method converges.* In the the first experimental setting of Biased-MNIST, we see that ReBias successfully converges; In the case of ImageNet however our results show that the model fails to converge. This failure of convergence is due to exploding gradients, moreover we observe the this can not be solved by tweaking hyperparameters.

- *Claim 2: The deliberately texture biased model $g$ helps $f$ reduce cross bias.* In case of Biased MNIST we see that the texture biased model $g$ indeed helps improve the cross-bias accuracy of the model $f$. However, it should be noted that this may fail to remove cross-biases from $f$ that are not textual in nature (for instance, shape biases or biases due to the existence of some object in the background). In such cases simply having low-receptive fields does not necessarily ensure that the model $g$ would be prone to texture biases more than the model $f$ would. Hence this method is more effective in improving a model's ($f$) cross-bias accuracy only when the cross-biases are caused by textures, similar to the Biased-MNIST setting.

- *Claim 3 : The Hilbert-Schmidt Independence Criterion (HSIC) utilized in the min-max objective function of ReBias is useful in reducing cross bias.* This claim seems to be satisfied by our Biased MNIST results. However the additional results we report even for sufficiently high $\rho \in [0.90, 0.99]$ show that the vanilla model itself outperforms the ReBias model. This may be due to the simplicity of the MNIST model setting.

- *Claim 4 : ReBias solves the cross-bias generalization problem.* From our experiments we observe that ReBias quite certainly is not able to fully solve the cross-bias generalization problem. This actually stems from the model's inadequacy to fully solve the requirements set by the previous claims, most significant of which is the failure to achieve convergence.

## 7   Conclusion and discussion

We conclude this reproducibility report by summarizing our results, clarifying the paper's implementation of methods ReBias is compared against, and commenting on the effectiveness of the experiments conducted in terms of the claims made by the paper.

**Our Results**

In our report we elucidate the technical details of ReBias, the theory behind it, and our attempts at replicating the results of this paper. We were successful in doing so for one synthetic dataset, Biased MNIST, and unsuccessful for another: 9 Class ImageNet. In addition, we provide analysis of the interpretability of the MNIST results, and the reproducibility challenges of ImageNet from the authors' implementation.

Although the authors clearly justify their intuition and theoretical motivation, we believe the empirical experiments conducted do not fully demonstrate the benefits of the proposed technique over existing methods. Our inability to achieve convergence in the 9-class ImageNet experiment, despite using the same training pipeline and hyper-parameters, is one such indicator of the inherent stochasticity involved in deep learning experiments leading to a difficulty in reproducing results.

**Clarifying implementation of methods ReBias is compared against**

Further, we are not able to compare ReBias's performance against HEX [3], this is primarily because the implementation of HEX on the problem settings discussed in this paper are not made public. Learned-mixin' [4], another ReBias is compared against in the original paper, uses an ensemble based approach to remove biases. However, its implementation, in the code linked to this paper, does not use such an approach. The implementation of RUBi [2] used in this paper (shown in Algorithm 4), is modified from a Visual-Question-Answering (VQA) setting to a Classification setting. However, as opposed to Algorithm 4, the original implementation doesn't utilize the Hilbert-Schmidt Independence Criterion, and has a different optimization strategy from that used by the authors in this work. The original optimization strategy of RUBi attempts to remove unimodal biases – *superficial correlations between inputs from one modality (say, the image input) and the answers without considering the other modality (the text input)* from VQA models; RUBi reduces the loss for examples that can be correctly answered without looking at the image while attempting to increase the loss for examples that cannot be answered without using both modalities. Naturally, the authors of ReBias had to make certain design choices for translating this idea to an image classification setting where there is just one modality of the input (an Image) to models (a target network $f$ and a texture biased model $g$).

**Final comments**

Attempting to reproduce the results presented in the ReBias paper gives us the opinion that, while ReBias proposes a novel training methodology, it's effectiveness for specific de-biasing use cases needs a more constructive empirical survey.

We would like to acknowledge the importance of the Reproducibility Challenge, that incentivizes members of the Machine Learning community to address and help repair the lack of standardization in this field.

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
