# OpenReview forum: "RC2020 Report: Learning De-biased Representations with Biased Representations"
_ML_Reproducibility_Challenge/2020 — Reject_

### Official Review · AnonReviewer1 · 2021-02-14
**Good effort but needs more experiments to assess reproducibility of the original work**

**Rating:** 4
**Confidence:** 4

**Review:**

**Reproducibility summary:** The summary is overall clear and well summarizes the authors' reproducibility effort.

**Scope of reproducibility:** The authors state four concrete claims they aim to verify. However, Claim 2 ("The deliberately texture biased model g also reduces cross bias") was confusingly stated, as *g* is a "biased" model from a bias characterizing model class, that is designed to suffer from more bias. I believe the authors meant to say "using *g* to train *f* helps reduce cross bias."

**Code:** The authors used the original authors' code with minor modifications. The authors made their version of the code available with appropriate documentation.

**Communication with the original authors**: The authors communicated with the original authors and received clarifications on the HEX, RuBi, and Learned-Mixin implementations, and WNIDs for constructing the 9-Class ImageNet dataset. However, I wish the authors would have done additional communication with the original authors on the following fronts: (1) Even after receiving clarifications on HEX and Learned-Mixin, the authors write that they were unable to compare them to ReBias. It would have been nice if the authors did further communications with the original authors to get HEX and Learned-Mixin (as well as StylisedImageNet, another work compared in the paper although the authors do not discuss it) working. (2) The authors discuss that they were unable to train ReBias on 9-Class ImageNet. As this is the main dataset of the paper, I wish the authors had reached out to the original authors to resolve the difficulties.

**Hyperparameter search:** The authors didn't conduct a hyperparameter search. However, they were forced to reduce the batch size from 128 to 16 due to their limited computational resources, and tried other learning rates and step sizes to counteract the smaller batch size.

**Ablation study**: The authors didn't conduct any ablation studies.

**Discussion on results:** On Biased MNIST, I'm confused by the way the authors presented their results because their results are not within 1% of the original results in the paper. I suggest the authors to double check this claim and include a side-by-side comparison with the original results. On 9-Class ImageNet, the authors state that they succesfully trained the Vanilla and Biased model, while they failed to train the ReBias model due to exploding gradients. Still I wish the authors had included their results in the report so that the readers can get a sense of how different the results are.

**Recommendations for reproducibility:** The authors don't provide explicit recommendations to the original authors. However, their descriptions of the difficulties they ran into (e.g. insufficient explanation of the implementation of prior works, insufficient explanation of the construction of 9-Class ImageNet, gradient explosion problem) may help the original authors improve the reproducibility of the work.

**Results beyond the paper**: For Biased MNIST, the authors conduct edadditional experiments with $\rho=0.98, 0.95, 0.9, 0.85$ and make an interesting observation that the Vanilla model beats ReBias for lower $\rho$, providing an enlarged picture. Furthermore, they posed two very interesting questions (bottom of page 6) although the authors did not attempt to answer them.

**Overall organization and clarity:** Overall, the report was organized and clearly written. However, there are several typos and grammatical errors that I hope the authors address in their revision.

**Minor comments:** (1) Figure 1 was helpful in understanding $\rho$ in the Biased MNIST experimental setup. (2) Tables 1-2 are unnecessary as Tables 3–4 subsume them. (3) Figure 2 was not very helpful in understanding the change in Vanilla/ReBias performance because the two lines are almost always on top of each other. (4) The authors made a typo in the learning objective for g in Algorithms 1, 3, and 4.

**Summary**: Overall, the authors made good effort to reproduce the original work and provided additional insights which I appreciate. The authors successfully reproduced Biased MNIST results and failed to reproduce 9-Class ImageNet results. However, because they do not report results for 9-Class ImageNet, which is the main dataset studied in the original paper, I found the evidence provided in this report insufficient for assessing the reproducibility of the original work. For the assessment, I suggest the authors to communicate with the original authors to resolve the difficulties with 9-Class ImageNet and/or try to reproduce action recognition results.

**Familiar With The Original Paper:**

I have read the original paper

**Reproducibility Summary:**

Report has summary

---

### Official Review · AnonReviewer2 · 2021-02-27
**A detailed report with meaningful analysis**

**Rating:** 6
**Confidence:** 4

**Review:**

*Problem statement:
The paper clearly states the reproducing details, together with the detailed results and analysis.

*Presentation:
The paper is well-organized and well-written.

*Communication with original authors:
The authors had some communication with the original authors.

*Code:
The code is available on GitHub and can be reproduced.

*Recommendations for reproducibility:
The authors provided useful comments for reproducing the original paper. I have read the code and found those comments are consistent with the provided codes.

*A few concerns
**Some minor typos:
** It will be even better if the authors can provide a simple illustration on the two major algorithms. The illustration would help others to follow this report.
** It will be better if the authors can provide a detailed readme or other instructions for the code. Currently, although the codes are well-organized, it still requires much time for users to debug some deatils.
** Some minor typos: Section 5.1.2, present -> presented

**Familiar With The Original Paper:**

I have read the original paper

**Reproducibility Summary:**

Report has summary

---

### Decision · Program_Chairs · 2021-03-31

**Decision:**

Reject

**Comment:**

Overall reviews and/or the paper content not good enough for the AC to recommend to the journal.